# Immunogenicity of the *Xcl1*-SARS-CoV-2 Spike Fusion DNA Vaccine for COVID-19

**DOI:** 10.3390/vaccines10030407

**Published:** 2022-03-08

**Authors:** Hailong Qi, Zhongjie Sun, Yanling Yao, Ligong Chen, Xuncheng Su

**Affiliations:** 1State Key Laboratory of Elemento-Organic Chemistry, College of Chemistry, Nankai University, Tianjin 300071, China; qihailong@newishes.com (H.Q.); szj@mail.nankai.edu.cn (Z.S.); 2Hebei Immune Cell Application Engineering Research Center, Baoding Newish Technology Co., Ltd./Newish Technology (Beijing) Co., Ltd., Beijing 100176, China; yaoyanling@newishes.com; 3School of Pharmaceutical Sciences, Key Laboratory of Bioorganic Phosphorus Chemistry and Chemical Biology (Ministry of Education), Tsinghua University, Beijing 100084, China

**Keywords:** SARS-CoV-2, Xcl1, DNA vaccine

## Abstract

SARS-CoV-2 spike (S) variants that may evade antibody-mediated immunity are emerging. Evidence shows that vaccines with a stronger immune response are still effective against mutant strains. Here, we report a targeted type 1 conventional dendritic (cDC1) cell strategy for improved COVID-19 vaccine design. cDC1 cells specifically express X-C motif chemokine receptor 1 (Xcr1), the only receptor for chemokine Xcl1. We fused the *S* gene sequence with the *Xcl1* gene to deliver the expressed S protein to cDC1 cells. Immunization with a plasmid encoding the S protein fused to Xcl1 showed stronger induction of antibody and antigen-specific T cell immune responses than immunization with the *S* plasmid alone in mice. The fusion gene-induced antibody also displayed more powerful SARS-CoV-2 wild-type virus and pseudovirus neutralizing activity. Xcl1 also increased long-lived antibody-secreting plasma cells in bone marrow. These preliminary results indicate that Xcl1 serves as a molecular adjuvant for the SARS-CoV-2 vaccine and that our Xcl1-S fusion DNA vaccine is a potential COVID-19 vaccine candidate for use in further translational studies.

## 1. Introduction

In December 2019, an outbreak of severe pneumonia was reported in Wuhan, China. Scientists identified the virus SARS-CoV-2 as the etiologic cause [1]. The SARS-CoV-2 infection that caused the pneumonia disease was named COVID-19 by the World Health Organization (WHO) on 11 March 2020, and it quickly became a global epidemic. Many treatment strategies have been developed to treat COVID-19 [2]. However, to recover prepandemic normalcy, a secure and efficient vaccine was still an urgent need. SARS-CoV-2, severe acute respiratory syndrome-associated coronavirus (SARS-CoV) and Middle East respiratory syndrome coronavirus (MERS-CoV) that have caused serious respiratory diseases in humans are all members of the Coronaviridae family. These RNA viruses go in a positive sense, are made of a single strand and are composed of four main structural proteins, spike (S), envelope (E), membrane (M), and nucleocapsid (N), and multiple nonstructural proteins [3,4,5]. The S protein of the SARS-CoV-2 binds to the cellular receptor angiotensin-converting enzyme 2 (ACE2), which mediates its entry into the host cell. Therefore, neutralizing antibodies and vaccines are aimed at the S protein and their effectiveness has been demonstrated in preclinical and clinical trials [4,6].

There are around 273 vaccine candidates for SARS-CoV-2 in the preclinical and clinical phases of development [7,8,9,10,11]. These vaccine candidates are based on inactivated virus, virus-like particle, DNA, mRNA and the recombinant protein vaccine platform. Among these vaccine strategies, synthetic DNA vaccine approaches have many advantages in comparison with other vaccine strategies. Designing DNA vaccines can be completed in a few weeks, which accelerates the development timelines. Moreover, DNA vaccines can induce a more intensive cellular immune response in comparison to inactivated virus and the recombinant protein vaccine. Compared to mRNA vaccines, DNA vaccines show temperature stability and do not need cold chain conditions, which was crucial in vaccine delivery in districts with limited resources. Plasmid DNA vaccines are also safe, easy to produce and scalable [12,13,14].

Naïve T cell responses and activation need conventional dendritic cells (cDCs) to phagocytose, process and present antigen. Appearing proofs reveal that the delivery of antigens to cDCs was an efficient method for inducing powerful T cell immune responses [15]. Moreover, targeting antigens to cDCs can improve antibody responses and was also observed in a number of previous studies [16]. Indeed, through specific antibody or ligands, binding to surface molecules or receptors such as MHC class II (MHC-II), CD11c, CD180, and Clec9A of cDCs for antigen delivery has induced strong antibody responses without adjuvant [17,18,19]. The plasmid DNA vaccine paved the way for genetically fusing antigens to antibodies or chemokine ligands to target surface markers on APCs. Previous studies have developed a number of DNA vaccines expressing fusion proteins targeting surface receptors on APCs, such as MHC-II, CCR5, CD40, CD14, and Xcr1 [20,21,22,23,24]. Immunization with these DNA vaccines enhances immune responses and induces protection against tumor and virus infection. Among these receptors, the Xcr1 receptor is specifically expressed on cDC1 cells in mice and humans. Targeting antigen to the Xcr1 receptor through its ligand Xcl1 increases both antibody and CD8^+^ T cell responses to produce protection in influenza and cancer models [24,25,26,27]. Emerging evidence suggests that targeting Xcr1 may be an approach to improve the antibody response for SARS-CoV-2 [28,29,30].

We generated a Xcl1-S fusion vaccine to evaluate Xcr1 targeting in a SARS-CoV-2 vaccine. We described the design and compared the results of the preclinical testing of Xcl1-S synthetic DNA vaccine candidates with the S-alone synthetic DNA vaccine. We showed similar expression of the fusion antigen and S antigen after the transfection of 293T cells in vitro. The expressed fusion antigen bound to cDC1 cells in vitro and homed to draining lymph nodes in vivo. After mice immunization with each antigen, serum and lung fluids from mice immunized with Xcl1-S fusion antigen had higher levels of specific antibodies and displayed enhanced live virus neutralization activity. The fusion vaccine also induced more potent CD8^+^ T cell immune responses. Our results suggest that the Xcl1-S DNA vaccine has good immunogenic potential against COVID-19, encouraging future translational studies to contain the current pandemic.

## 2. Materials and Methods

### 2.1. Cell Lines and Plasmids

We purchased HEK-293T (ATCC^®^ CRL-3216™) cell lines from ATCC. HEK293T cells that stably express ACE2 were obtained from OBiO Technology Company (#HYC3694). Cells were preserved in DMEM with 10% fetal bovine serum (FBS) and penicillin-streptomycin before assay. The Xcl1-S and S gene sequences were synthesized and cloned into the pVAX1 vector by TSINGKE Biological Technology.

### 2.2. Western Blot Assay

Whole cellular or tissue extracts (WCE) were prepared in RIPA buffer (50 mM Tris-HCl, pH 8.0, 150 mM NaCl, 1.5 mM MgCl_2_, 0.1% SDS, 0.5% deoxycholate (DOC), 1% NP-40, 1 mM PMSF, 1× protease inhibitor mixture (Roche), 1× phosphatase inhibitor mixture, 1 μg/mL aprotinin, 1 μg/mL leupeptin, and 1 μg/mL pepstatin). SDS-PAGE loading buffer (5×) was added to whole cellular extracts and boiled, followed by separation on SDS-PAGE gels and blotting using the anti-FLAG antibody (Sigma #1804, Burlington, MA, USA).

### 2.3. Animals and In Vivo Vaccine Immunization

Eight-week-old male C57BL/6 mice were acquired from Charles River Laboratories. The mice were raised at Peking Union-Genius Pharmaceutical Technology Company (Beijing, China). All tests were conducted following the ethical regulations and received approval from the Peking Union-Genius Institutional Animal Care and Use Committees (IACUC). The ethical approval number for this study is JY-21002. Starting from day 0, mice received 25 µg of plasmid DNA injected into the tibialis anterior (TA) muscle, and subsequently the animals were immediately subjected to TERESA in vivo electroporation (EP) to promote plasmid entering into the muscle cells. The parameters for TERESA EP delivery process are 60 V voltage, 10 Hz frequency and 50 ms interval. Peripheral blood was subjected to S-specific antibody detection on days 0, 14, 21, 28 and 35. Mice from each group were euthanized with CO_2_ after 28 days of the immunization to analyze the cellular immune response and the antibodies in the bronchoalveolar lavage (BAL) fluid.

### 2.4. Antigen Binding ELISA

S-protein-specific serum antibody titers were measured using ELISA. Corning ELISA plates were incubated with 10 µg/mL of the recombinant S protein purified in our lab in phosphate-buffered saline (PBS) overnight at 4 °C. After three washes, the plates were blocked with 1 × ELISAPOT Diluent (Invitrogen, cat. 00-4202-56, Waltham, MA, USA) for 2 h at 25 °C. Following another washing, the plates were incubated overnight using mouse serum at 4 °C and diluted with PBST. Following a set of washings, the plates were incubated with a 1:5000 dilution of goat anti-mouse IgG (H + L)-BIOT, IgG1-HRP, IgG2a-HRP or IgG3-HRP (Southern Biotech, cat. 1036-08, 1071-05, 1081-05 and 1101-05) for 1 h at room temperature. For IgG (H + L)-BIOT, following three washes, the plates were incubated with streptavidin-HRP (Solarbio, cat. SE068, Beijing, China) for 30 min at 25 °C. Finally, after washing the plates, the reaction was revealed using TMB 1-Component Peroxidase Substrate (Invitrogen, cat. 00-4201-56) and impeded using a 2 M HCl solution. The absorbance at 450 nm was determined within 30 min using a Synergy HTX instrument (BioTek Instruments, Highland Park, VT, USA).

### 2.5. IFN-γ and IL-4 ELISPOTs

The spleen was cut into pieces and ground to disperse into single cell. 1XRBC lysis buffer (Biolegend, cat. 420301, San Diego, CA, USA) was added to remove red blood cells. Splenocytes were counted and 2 × 10^5^ splenocytes per well were plated into anti-IFN-γ or anti-IL-4 plates and restimulated with the N terminal domain (NTD), receptor binding domain (RBD) and S1 peptides pools with 5 µg/mL concentration for 18 h at 37 °C 5% CO_2_ (Sino Biological, cat. PP001, PP002 and PP003-A, Beijing, China). The magnitudes of IFN-γ or IL-4 secretion were evaluated in accordance with the manufacturer’s protocol (Mabtech, cat.3321-4APW-2, cat. 3311-4APW-2, Nacka Strand, Sweden).

### 2.6. B Cell ELISPOTs

Recombinant wild-type, delta and omicron S protein (Sino Biological, cat. 40589-V08B1, 40589-V08B16 and 40589-V08H26) were dissolved with PBS and coated into ELISPOT plates (MabTech, cat. 3654-WP-10) at a concentration of 2 µg per well. The tibiae were dissected from vaccinated mice and bone marrow was flushed out with PBS following red blood cells lysis. The cell suspension was filtered through a 70 µm Nylon cell strainer and was added to ELISPOT plates at a concentration of 5 × 10^5^ cells per well for 20 h at 37 °C 5% CO_2_. Plates were washed 5 times with PBS and incubated with Goat anti-Mouse IgG ALP (A2429; Sigma) for 1 h. After washing 5 times, the plates were developed with BCIP/NBT-Purple liquid substrate system for membranes (B3679; Sigma).

### 2.7. Flow Cytometry

After the spleens were grinded into single cells and red blood cells were lysed, CD11c^+^ DCs were enriched with CD11c Microbeads UltraPure (Miltenyi Biotech, cat. 130-108-338, Cambridge, MA, USA). 2 × 10^5^ CD11c^+^ DCs were incubated with 600 µg Flag-tagged Xcl1-S, S or vector plasmid transfected 293T cell lysate for 30 min. Then, the cells were washed two times with PBS and subjected to FITC anti-Flag antibody (Abcam, cat. ab117505) for 30 min measured by flow cytometry using a FACS CANTO TWO flow cytometer (BD Biosciences). 1 × 10^6^ C57BL/6 mice splenocytes were plated on a 24-well plate supplied with overlapping peptides covering the SARS-CoV-2 S protein for 18 h at 37 °C in 5% CO_2_ to stimulate T cell. Mice splenocytes were collected, and intracellular cytokine levels were measured by flow cytometry using a FACS CANTO TWO flow cytometer (BD Biosciences) after permeabilization and fixation of the cells (BD Biosciences). Cells were stained using the following antibodies from Biolegend: FITC anti-mouse CD8a, PE anti-mouse CD4 and APC anti-mouse IFN-γ. Finally, the results were examined with FlowJo software (FlowJo LLC, Ashland, OR, USA).

### 2.8. SARS-CoV-2 Pseudovirus Neutralization Activity Tests

SARS-CoV-2 pseudoviruses were generated by transfection of pcDNA3.1 (Invitrogen) expression vectors encoding the respective spike proteins into 293T cells using Lipofectamine 3000 (Invitrogen, cat. L3000015). The transfected cells were infected with mCherry VSV-G 24 h after transfection. New culture medium was added 2 h after infection, and then viral supernatants were collected 48 h later. Neutralization assays were performed by incubating pseudoviruses with serial dilutions of serum at 37 °C for 90 min. Xcl1-S vaccinated and control mice serums were treated for 30 min at 56 °C before dilution. Following the incubation period, the mixture was introduced into HEK293T cells that stably express ACE2 (OBiO Tech-nology Company #HYC3694, Shanghai, China) and further incubated at 37% humidity and 5% CO_2_ for 48 h. Half-maximal neutralization concentrations (NT50) of the evaluated serum antibodies were determined by mCherry fluorescence inhibition rate 48 h after exposure to virus–srum mixture using GraphPad Prism 7 (GraphPad Software, Version 7.00).

### 2.9. SARS-CoV-2 Wild-Type Virus Neutralization Activity Tests

SARS-CoV-2 wild-type virus neutralization assays were done at the Wuhan Institute of Virology, Chinese Academy of Sciences (CAS), and authorized by the National Health Commission of the People’s Republic of China. Serums obtained from the inoculated mice were heat inactivated at 56 °C for 30 min. The mouse serum was diluted with MEM medium (NZK Biotech) containing 2% FBS (Gibco, cat. 10100147, Shanghai, China) at 1:5, and then diluted to 1:640 in 2-fold increments, with 8 dilutions for each serum. Then, 100 µL 2 × 10^3^ TCID50/mL SARS-CoV-2 viruses (National virus resource bank IVCAS 6.7512; BetaCoV/Wuhan/WIV04/2019) was mixed with the diluted serum in equal volume and placed at 37 °C for 1 h. The 200 µL virus and serum mixture was added into 96-well plates paved with 10^4^ per well Vero-E6 cells (ATCC). After being cultured at 37 °C and 5% CO_2_ for 72 h, cells were fixed with 4% paraformaldehyde (HuShi, cat. 30525-89-4, Guangzhou, China) for 1 h. 0.2% TritonX-100 (Sigma, cat.9002-93-1) was added to the inactivated cells for permeabilization for 10 min and blocked with 5% BSA at 37 °C for 1 h. The rabbit polyclonal antibody against SARS-CoV-2 N protein (prepared in the laboratory) was incubated at 37 °C for 2 h, and then Sheep anti rabbit IgG labeled with Alexa fluor^®^ 488 fluorescent was incubated at 37 °C for 1 h. Finally, after being treated with Hoechst 33258 (Beyotime, cat, C1018, Nantong, Jiangsu, China) dye for 10 min, the fluorescence was scanned by high connotation cell analyzer.

### 2.10. Statistics

For the statistical analyses, GraphPad Prism 7 software (La Jolla, CA, USA) was utilized. Two-tailed t-tests were used to compare differences among groups, which were considered statistically significant if *p* < 0.05. In each graphic, mean values are represented by lines, and the standard errors of the means are represented by error bars.

## 3. Results

### 3.1. Design and Synthesis of Xcl1-SARS-CoV-2 S Gene Fusion Constructs

As a method to specifically deliver the S protein to cDC1 cells, we used a linker containing 11 amino acids, (glycine)5-serine-(glycine)5, to fuse the mouse Xcl1 C-terminus to the S protein N-terminus and generate the mXcl1-S protein. To ensure the fusion protein was secreted, we deleted the transmembrane domain of the S protein (amino acids 1214–1234). We also deleted the signal peptide (amino acids 1–12) from the S protein to avoid a breakage between Xcl1 and the S protein. A 3xFlag tag coding sequence was added to the carboxy-terminal region to better identify the fusion protein expression and distribution in vitro and in vivo. The fusion protein was made by cloning the selected DNA sequence, which was synthesized and cloned into the pVAX1 expression vector (under the control of the human cytomegalovirus promoter (CMV) and a bovine growth hormone (BGH) polyadenylation signal) (Figure 1A). The fusion protein has 1377 amino acids and weights approximately 180 kDa. We analyze the 3D structural model of the fusion proteins at the online structure prediction server (https://zhanggroup.org/, accessed on 20 July 2021). The chemokine XCL1 retains its conformation in the fused protein (Figure 1B) [31].

### 3.2. Identification of the Expression of the DNA Vaccine Constructs In Vitro and In Vivo

We then transfected HEK293T cells and analyzed the expression of the Xcl1-S fusion protein 24 h after transfection. We conducted Western blot analysis with a Flag antibody and found that the fusion protein migrated at 180 kDa, consistent with the predicted molecular weight (Figure 2A). We also detected the expression and lymph node targeting ability of the Xcl1-S fusion proteins in vivo. Three days post the intramuscular injection and electroporation, we resected and homogenized the muscles and inguinal lymph nodes for Western blot assays. The expression of Xcl1-S was indistinguishable from that of the S protein in the muscle, but was significantly increased in the lymph node (Figure 2B). We next enriched the DCs with a CD11c microbeads from the mice spleen and conducted an in vitro binding assay on spleen DCs using the Xcl1-S plasmid transfected cell lysate to validate the activity of Xcl1 in the expressed fusion protein. Notably, Xcl1-S-transfected cell lysates showed a more than 3-fold enhanced interaction with DCs (CD11c^+^) compared with the same amount of S-transfected cell lysate (Figure 2C). Based on these results Xcl1-S effectively serves as a chemoattractant to the lymph node and targets to cDC1 cells.

### 3.3. Vaccination with the Xcl1-S Plasmid Generates Stronger S-Specific Humoral Immune Responses in C57BL/6 Mice than the S Plasmid

C57BL/6 mice were inoculated with 25 µg of plasmid encoding Xcl1-S, S, or pVAX1 at a 2-week interval to test the ability of Xcl1-S to induce a protective immune response. Two immunizations were conducted, and peripheral blood was collected at the indicated times for the ELISA assay of recombinant S-protein-specific antibodies (Figure 3A,B). Both Xcl1-S and S showed almost equivalent S-specific antibody responses at days 14 and 21. However, the antibody titer of Xcl1-S-immunized mice continued to increase two weeks after the second vaccination compared to that of S-immunized mice (Figure 3B). We collected the serum from day 35 and separately measured the serum IgG binding endpoint titers against recombinant SARS-CoV-2 spike protein RBD, S1 and S1 + S2 ECD regions. Fusion with Xcl1 increased the endpoint titers by 2–4 times compared to S-immunized mice (Figure 3C–E). We next analyzed the antibody subclass. Serum samples from day 35 were evaluated for S-specific IgG1, IgG2a, and IgG3. Xcl1-S immunization induced similar levels of IgG1. However, IgG2a and IgG3 were significantly increased in Xcl1-S mice (Figure 3F). We next evaluated whether fusion with Xcl1 could increase long-lived antibody-secreting plasma cells in bone marrow. BM was harvested after 16 weeks from the second vaccination for detecting SARS-CoV-2 wild-type, delta and omicron S-specific antibody-secreting cells (ASC) through ELISPOT analysis. We found that Xcl1-S-immunized mice induced significantly higher numbers of ASC in BM 16 weeks after immunization compared with S-immunized mice (Figure 3G). Moreover, ASC from Xcl1-S-immunized mice showed an almost equal response to delta and a slightly low response to omicron S protein (Figure 3G).

### 3.4. The Serum of Xcl1-S-Immunized Mice Showed High Levels of Neutralization Activity

We next sent out to assess the neutralization effect of serum derived from mice immunized with Xcl1-S, S or pVAX1 for SARS-CoV-2 pseudovirus and wild-type virus. We have developed a neutralization assay with a VSV pseudovirus system, and the backbone was provided by VSV G pseudotyped virus that packages expression cassettes for mCherry instead of VSV-G in the VSV genome based on a previous study [32,33]. This VSV-based pseudovirus displayed the SARS-CoV-2 spike protein. Neutralization titers were detected by a reduction in the number of mCherry expression cells compared with the no-serum positive controls. The pseudoviruses that carry the mCherry gene were incubated with serum at 37° for 90 min and transferred to infect stable expression of ACE2 HEK293T cells. The mCherry positive cells were counted and the inhibition was calculated after 48 h of culturing. Although Xcl1-S and S serum showed no significant neutralization activity when these samples were assessed against the pseudovirus due to large individual variation in group, the average neutralization titer of Xcl1-S was five times higher than that of S (Figure 4A,B). Moreover, Xcl1-S sera displayed higher neutralization titers against the wild-type virus (Figure 4C,D). These results provide evidence for Xcl1-S as a better vaccine than S and support the usage of Xcl1 as a molecular adjuvant for SARS-CoV-2 vaccine elaboration.

### 3.5. Detection of the Distribution of SARS-CoV-2-Specific Antibodies in the Lung

As SARS-CoV-2 invades the human body through the respiratory tract, effective distribution of specific antibodies in the lung mucosa could protect against lower respiratory disease (LRD), which could lead to severe cases for COVID-19. Hence, we collected alveolar lavage fluid and measured the S-protein-specific antibodies in the lungs. We immunized C57BL/6 mice on days 0 and 14 with 25 µg Xcl1-S, S or pVAX1 control plasmid. We sacrificed the mice on day 28, and about 0.5 mL of alveolar lavage fluid for each mouse was obtained and analyzed with an ELISA containing the SARS-CoV-2 S protein. Mice that received the Xcl1-S plasmid had significantly higher amounts of specific S protein antibodies in alveolar lavage fluid than mice inoculated with S and control plasmids (Figure 4C). Altogether, our results show that immunization with Xcl1-S could induce specific antibodies against SARS-CoV-2 present in the lungs.

### 3.6. Vaccination with the Xcl1-S Plasmid Generates Stronger S-Specific Cellular Immune Responses in C57BL/6 Mice than the S Plasmid

Previous studies suggest that the cellular immune response regulates the SARS-CoV-2 infection [34,35]. We sacrificed the mice after two weeks from the second vaccination and collected spleen lymphocytes for IFN-γ and IL-4 ELISpot assay. Splenocytes were stimulated for 18 h with pools of 15-mer overlapping peptides spanning the NTD, RBD and S1 of the SARS-CoV-2 S protein. Both S and Xcl1-S-immunized mouse splenocytes are activated in response to the three-peptide-pool stimulation. The S1 peptide pool leads to the highest-magnitude IFN-γ secretion. Fusion with Xcl1 significantly increased the spot numbers. However, there were no IL-4 response spots observed in both S and Xcl1-S groups, indicating a Th1 bias response (Figure 5A,B). The proportion of CD8^+^ cells secreting IFN-γ was significantly higher in the Xcl1-S-immunized mice than in the S-immunized and unimmunized mice. No differences in IFN-γ secretion by CD4^+^ cells were observed (Figure 5C,D). Taken together, these results suggest that Xcl1-S vaccination-cellular immune responses are more intensive.

## 4. Discussion

This study targeted the SARS-CoV-2 S protein to cDC1 using the receptor Xcr1, a specific marker of cDC1 cells, by fusing with its ligand Xcl1. The Xcl1-S fusion protein specifically binds to DCs and enhances the antigen-specific immune response. In a SARS-CoV-2 pseudovirus and wild-type virus neutralization assay, the Xcl1-S fusion vaccine enhanced the neutralization of the wild-type virus and induced long-lived antibody-secreting cells in bone marrow, suggesting that this might be a promising vaccine approach for SARS-CoV-2. Xcl1-S fusion vaccination also produced protective CD8^+^ T-cell responses, indicating that Xcl1 could be a molecular adjuvant for the SARS-CoV-2 vaccine design. A previous study found that deletion of the first amino acid prevented receptor-mediated endocytosis and enhanced protective antibody responses against influenza virus [25]. However, fusion of the S protein with full-length mouse Xcl1 significantly increased both the cellular response and neutralization titer of the antibody in our study, which may be caused by the difference in the delivered antigens. The cellular and molecular mechanisms underlying this different response remain to be established. Moreover, whether the fusion of Xcl1 with the viral antigen could cause side effects should be evaluated in future studies.

It is also noteworthy that the S plasmid immunization-induced immune responses were lower than the literature [3]. One possible explanation is that we adopted different sequence optimization methods resulting in the differences in protein expression. On the other hand, we also did not use double prolines substitution or six prolines substitution to elevate the stability of S protein, which is a flaw of design in our study [36,37]. However, the fusion of Xcl1 did enhance the immune response of the S protein confirmed through the DNA vaccine platform, which could also be applied to mRNA, recombinant protein and other vaccine platforms to improve the efficacy of the SARS-CoV-2 vaccines. In our study, 25 µg DNA plasmid was used for evaluating immune response. Whether this dose can be applied to human body in equal proportion needs further confirmation. Actually, we found that the current clinical dosage of DNA vaccine is as high as 6 mg per dose without adverse effects [38,39,40]. The scale magnifying dose of 25 µg is about 8 mg to be administrated to a 60 kg human. Considering the good safety of the DNA vaccine in clinical trials, we believe that an 8 mg dose would not cause adverse effects in humans, which needs to be well evaluated.

Many SARS-CoV-2 variants have emerged since the discovery of this virus, such as SARS-CoV-2 B.1.1.7 that appeared in Southeast England and another SARS-CoV-2 strain B.1.351, which was identified in South Africa. Recently, the omicron strain spread globally. These new strains have become the dominant variants in the world. These new viral variants harbored different kinds of mutations distributed in the different domain of the S protein. Moreover, mutations that occurred in the ACE2 receptor-binding motif (RBM) often lead to an attenuated effect of vaccines or monoclonal antibody therapies. Increasing evidence indicates that individuals are more susceptible to get infected with a variant containing a mutation in the RBM if they have a weak antibody response to vaccination or it is their first infection. A recent evaluation of the protective efficacy of the available SARS-CoV-2 vaccines against the new variants revealed approximately 2 times less neutralizing activity against B.1.1.7 and about 6.5 to 8.6 times less activity against B.1.351 [41,42,43,44,45]. However, the third immunization of mRNA-1273 increased omicron neutralization titers, which indicates that enhanced vaccine-induced antibody and cell immune response could strengthen protection against the virus of concern (VOC) [46]. Emerging new SARS-CoV-2 drug-resistant strains have increased the urgency for developing potent and easily updated vaccines. Fusion with Xcl1 may help to develop more powerful vaccines in different platforms.

We are happy to see that the number of new infections has begun to decline following vaccination. However, many scientists expect SARS-CoV-2 to become endemic. The inclusion of Xcl1 as an immune enhancer to achieve more potent and easily improved vaccines will help in epidemic normalization and prevention.

## Figures and Tables

**Figure 1 vaccines-10-00407-f001:**
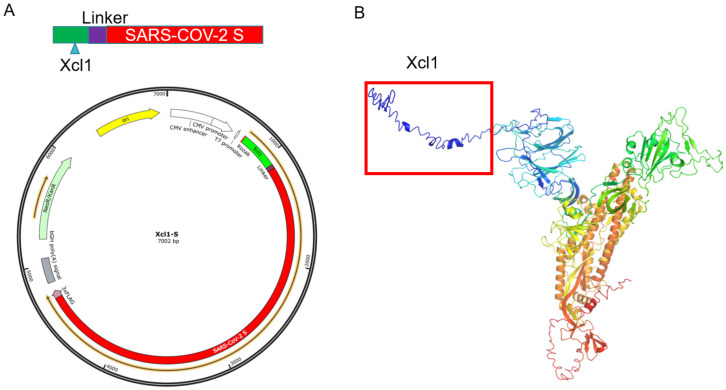
Design and prediction of the structure of Xcl1-S synthetic DNA vaccine constructs. (**A**) Schematic diagram of the Xcl1-S synthetic DNA vaccine construct. The glycine(5)–serine–glycine(5) linker sequence was inserted between the Xcl1 C-terminus and SARS-CoV-2 spike protein N-terminus. (**B**) Xcl1 maintains its conformation in the fusion protein, as analyzed with the online structure prediction server (https://zhanggroup.org/, accessed on 20 July 2021).

**Figure 2 vaccines-10-00407-f002:**
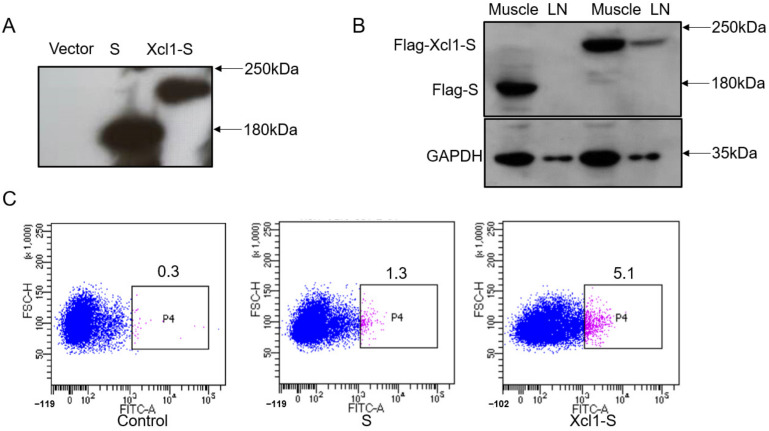
Identification of the expression and function of the DNA vaccine constructs. (**A**) Flag-tagged Xcl1-S and S were overexpressed in HEK293T cells and analyzed by immunoblotting using antibodies against Flag to indicate Xcl1-S and S, respectively. (**B**) Immunoblots confirming the distribution of Xcl1-S and in muscle and lymph nodes (LN) from plasmid-injected mice. Protein extracts from the lymph nodes and muscles of one representative pair of mice were tested with an anti-Flag antibody. (**C**) Binding of Flag-tagged Xcl1-S and S proteins to DCs. DCs isolated from the mouse spleen with CD11c beads were incubated with lysates from 293T cells expressing the Xcl1-S and S proteins.

**Figure 3 vaccines-10-00407-f003:**
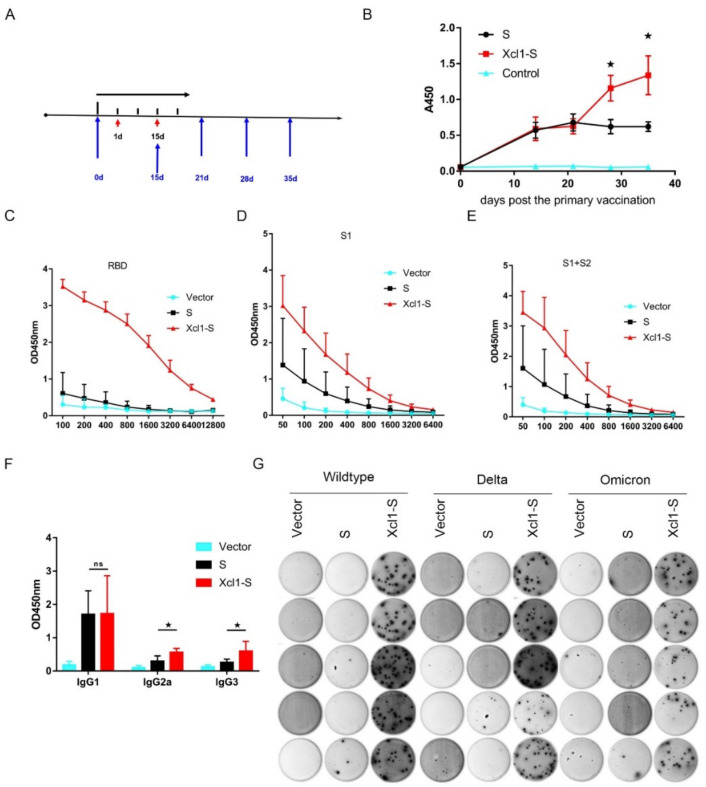
Xcl1-S vaccination elicited a significant increase in humoral immune responses. (**A**,**B**) C57BL/6 mice were immunized on days 0 and 14 with 25 µg of the Xcl1-S, S or pVAX1 plasmid. Mouse serum with a 1:100 dilution for IgG ELISA assay were collected at the indicated times. The data shown here represent OD450 nm values (means ± SEM) for each group of mice. (**C**–**E**) Mouse serum was collected 21 days after the second vaccination and were sequentially diluted at different concentrations for IgGs against the RBD, S1 and S1 + S2 ECD of the S protein using ELISA. Data are means ± SEM. (**F**) The IgG1, IgG2a and IgG3 isotypes of anti-S were examined in serum with a 1:100 dilution collected after the second immunization. (**G**) 5 × 10^5^ cells collected from bone marrow 16 weeks after the second vaccination were re-stimulated with wild-type, delta or omicron S protein for measuring S-specific long-lasting antibody-secreting cells. ★, *p* < 0.05.

**Figure 4 vaccines-10-00407-f004:**
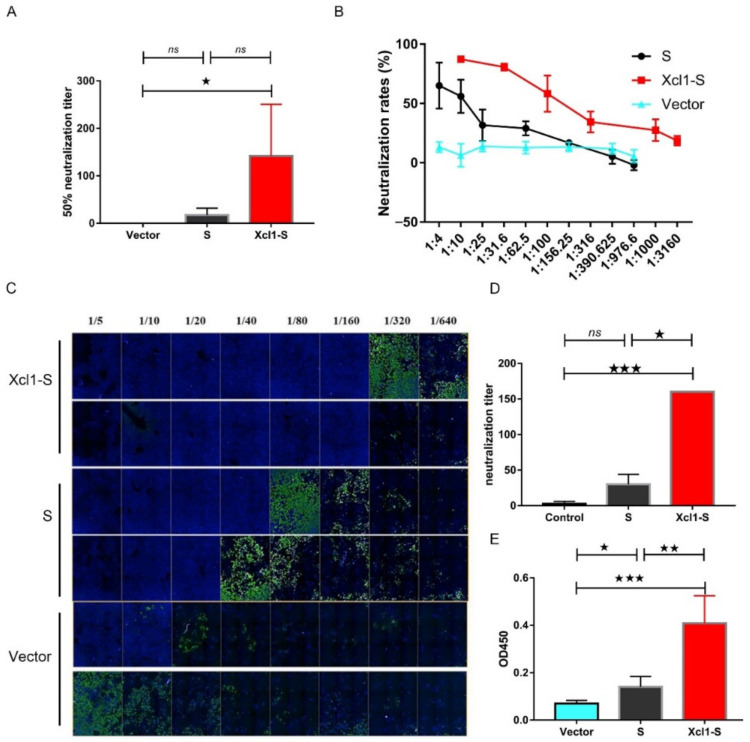
SARS-CoV-2 wild-type and pseudovirus neutralization assay. (**A**) The 50% neutralization titer of serum samples obtained from Xcl1-S-, S- or pVAX1-immunized mice were analyzed using VSV-based pseudovirus displaying spike, which carried an mCherry gene. (**B**) The mCherry fluorescence inhibition rates for different serum dilution ratios were displayed. Curve fitting was used to determine the half-maximal inhibitory concentrations (NT50), which were presented as the neutralization titers in (**A**). (**C**) Fold change in serum neutralization titers against wild-type viruses. (**D**) A statistical analysis was performed for (**C**) using a two-tailed paired *t* test. (**E**) Detection of SARS-CoV-2 S protein-reactive antibodies in alveolar lavage fluid. C57BL/6 mice (*n* = 5 per group) were immunized on days 0 and 14 with Xcl1-S, S or pVAX1. Alveolar lavage fluid was collected with a 1:100 dilution on day 28 and assayed for SARS-CoV-2 spike protein-specific IgG antibodies using an ELISA assay. ★, *p* < 0.05, ★★, *p* < 0.01, ★★★, *p* < 0.001.

**Figure 5 vaccines-10-00407-f005:**
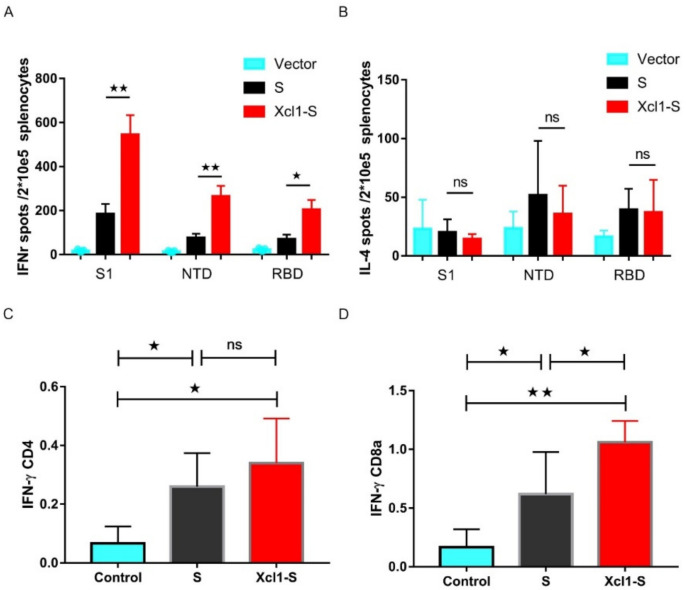
Xcl1-S vaccination elicited a significant increase in cellular immune responses. (**A**,**B**) The number of IFN-γ-expressing cells (**A**) and IL-4-expressing cells (**B**) from splenocytes were measured by ELISpot. Data are mean ± SEM of the spleens from *n* = 5 mice per group. (**C**,**D**) Splenocytes were stimulated with a combined mixture of S peptide pools for 18 h. The frequency of S-specific IFN-γ CD4^+^ and CD8^+^ T cells was determined using a flow cytometry analysis. Summary graphs showing the frequencies of IFN-γ CD4^+^ and CD8^+^ T cells after vaccination. Data shown in the graphs represent the average of five mice in each group, and error bars represent SEM. ★, *p* < 0.05, ★★, *p* < 0.01.

## Data Availability

Not applicable.

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
