# Peer review of "Immunogenicity of the Xcl1-SARS-CoV-2 Spike Fusion DNA Vaccine for COVID-19"

_vaccines, 2022, doi:10.3390/vaccines10030407_

Round 1

Reviewer 1 Report

This paper demonstrates successfully the use of a SARs CoV-2 spike fusion with XclI as a potential vaccine to counter Covid-19.  The fusion protein stimulates good levels of antibody and cell mediated responses in a mouse model.  The protein is delivered to cells via a DNA format, which has been proven as a good vaccine delivery system with other studies and application in humans.

The work is well presented and I have only a few comments:

  1. In Figure 1, the quality of the plasmid map is very poor, but this may just be a reflection of how it appears on my screen rather than the final published figure.  
  2. In Figure 2, the S protein is labelled as 180kDa, rather than the fusion product.
  3. The amount of DNA administered to mice seems very high at 25ug.  If this is scaled up to a human doses based on weight differential I can't see how this can be practical?  Could the authors comment on this?

Reviewer 2 Report

The study compared Xcl1-S and S plasmid immunizations in mice and found that compared to S, Xcl1-S induced stronger and longer antibody and T cell responses. Overall, the finding is significant. However, better descriptions of methods, especially the neutralization assays, are needed. Also, there should be Discussions on the flaws of study design, i.e. not using S2P or S6P for secreted S protein expression.

  1. 2.2. Western blot assay. To indicate secreted S and Xcl1-S expressions, shouldn’t the culture supernatant be used instead of cell lysate? Also, was S2P or S6P (PMID: 32703906) used for secreted spike protein expression? Note that both COVID-19 mRNA vaccines (Moderna and Pfizer) used S2P. This may explain the poor antibody response from the S plasmid immunization, a point that needs to be discussed. 
  2. 2.3. Animals and in vivo vaccine immunization. How much plasmid was used for TERESA electroporation? Also 25 ug?
  3. 2.4. Antigen binding ELISA. Where did the “recombinant S protein” come from? From Sino Biological? The source needs to be indicated.
  4. 2.5. IFN-γ ELISPOTs. What about IL-4 ELISPOT? Not described.
  5. 2.7. Flow cytometry. Figure 2C flow cytometry method was not described.
  6. 2.8. SARS-CoV-2 pseudovirus neutralization activity tests. What backbone was used to generate the pseudovirus? The authors mentioned “GFP” - how was GFP incorporated? Quantitative SARS-CoV-2 pseudovirus neutralization assays have been well established: PMID: 32454513; PMID: 32692348. Why did the authors choose such a poor/not-quantitative neutralization assay (poor results in Figure 4A)?
  7. 2.9. SARS-CoV-2 wild-type virus neutralization activity tests. What was the readout for this assay? “After 3 days, the plates were studied under the microscope” – what was studied under the microscope? Again, poor results in Figure 4B. “The statistical analysis was performed using a two-tailed paired t test.” – was there statistical analysis for Figure 4B?
  8. Figure 3B did not specify the antigen (S protein?). There was no figure legend for Figure 3G.

Author Response

Thank you for your positive decision that allows us to revise our manuscript. Please see the attachment.
